# Breast Abscesses Caused by Anaerobic Microorganisms: Clinical and Microbiological Characteristics

**DOI:** 10.3390/antibiotics9060341

**Published:** 2020-06-18

**Authors:** Fernando Cobo, Vicente Guillot, José María Navarro-Marí

**Affiliations:** Department of Microbiology and Instituto de Investigación Biosanitaria ibs.GRANADA, University Hospital Virgen de las Nieves, 18014 Granada, Spain; vicentel.guillot.sspa@juntadeandalucia.es (V.G.); josem.navarro.sspa@juntadeandalucia.es (J.M.N.-M.)

**Keywords:** anaerobes, breast abscesses, MALDI-TOF MS, antibiotics, bacteria

## Abstract

The objectives of this study were to report the antimicrobial susceptibility of 35 clinically significant anaerobic bacteria isolated from breast abscesses between March 2017 and February 2020 in a tertiary hospital in Granada (Spain) and to describe key clinical features of the patients. Species identification was performed mainly by MALDI-TOF MS. Antimicrobial susceptibility tests were carried out against benzylpenicillin, amoxicillin–clavulanic acid, imipenem, moxifloxacin, clindamycin, metronidazole, and piperacillin–tazobactam using the gradient diffusion technique and European Committee on Antimicrobial Susceptibility Testing EUCAST breakpoints (except for moxifloxacin). The most frequent anaerobes were *Finegoldia magna* (31.4%; n = 11), *Actinomyces* spp. (17.1%; n = 6), *Propionibacterium* spp. (17.1%; n = 6), and *Prevotella* spp. (14.2%; n = 5). Imipenem, amoxicillin–clavulanic acid, and piperacillin–tazobactam were universally active against all genera tested. High overall resistance rates to clindamycin were observed, especially for Gram-positive anaerobic cocci (56.2%) and Gram-positive anaerobic bacilli (38.4%). High resistance rates to metronidazole were also observed for Gram-positive (76.9%) and Gram-negative anaerobic bacilli (50%). High resistance rates to moxifloxacin were found for Gram-negative anaerobic bacilli (50%) and Gram-positive anaerobic cocci (31.2%). No breast abscess cases of *Bacteroides* spp. were detected. Routine antimicrobial susceptibility testing for anaerobes in breast abscesses may contribute to allow empirical therapies to be selected in accordance with local data on resistant strains.

## 1. Introduction

Breast abscesses constitute a clinical problem with a low overall incidence, and the majority of patients are managed in a community setting, mainly with antibiotics. They are localized infections caused by both aerobic and anaerobic microorganisms [1]. This clinical entity usually occurs in women during lactation, and they have been traditionally considered a minor clinical problem. Non-puerperal breast abscesses, however, have been uncommonly described in the medical literature and may be predisposed to by smoking and commonly relapse [2]. These two types of breast abscesses could have different presentations, risk factors, microbial agents, and treatments. 

The microbial etiology of breast abscesses has been previously published in different studies, and the main causative agents belong to the genera *Staphylococcus* [3,4,5], although other kinds of microorganisms have been implicated, such as *Streptococcus*, *Enterococcus*, and *Enterobacteriaceae*. However, in non-puerperal abscesses, a greater variability of microorganisms was found displaying mixed growth patterns [6]. Regarding the anaerobic pathogens, in some locations, these microorganisms cause nearly a third of cases of breast abscesses, especially *Bacteroides fragilis* [7], and they can be the etiological agents both of mixed infections and those isolated in pure culture.

The optimal treatment of breast abscesses remains a controversial issue, but recent evidence supports the need for a drainage procedure or an image-guided percutaneous aspiration. Obtaining an appropriate sample is a guarantee for a correct etiological diagnosis. Until now, few studies have focused on cases of breast abscesses due to anaerobes. In the present report, we analyze the clinical data and antimicrobial susceptibility of all breast abscesses produced only by anaerobes in our hospital over the last few years.

## 2. Results

### 2.1. Clinical Characteristics of Patients

The study included 24 cases of breast abscesses from 22 women. In two patients, a relapse of the disease was observed. The mean age was 44 years (range 27–88). Twenty cases (90.9%) were produced in non-puerperal women. Table 1 exhibits their main clinical characteristics.

### 2.2. Isolated Bacteria

The study included 35 non-repetitive clinically relevant anaerobic strains obtained from 24 samples. To distinguish true pathogens and contaminants, microbiological and clinical aspects were taken into account. The presence of abscess infection was defined as an episode of erythematous pain, localized edema with tenderness on palpation, and/or signs of suppuration in any breast location [4]. Patients with a history of at least two recurrent episodes and/or presentation with draining sinuses were also included [4]. By contrast, puerperal breast abscesses were defined as breast abscesses during pregnancy, during the first 3 months after labor without lactation, during lactation, or during the first 3 months after cessation of lactation [5]. From a microbiological point of view, isolates in pure culture with abundant growth in the plates were included. Additionally, mixed cultures with the presence of abundant polymorphonuclear leukocytes were considered. Fourteen strains (40%) were isolated in pure culture: *Finegoldia magna* (n = 7), *Prevotella* spp. (n = 2), *Peptoniphilus harei* (n = 2), *Propionibacterium acnes* (n = 2), and *Actinomyces turicensis* (n = 1). By contrast, 21 strains (60%) from 10 samples were isolated as mixed cultures: *Actinomyces* spp. (n = 5), *F. magna* (n = 4), *Propionibacterium* spp. (n = 4), *Prevotella* spp. (n = 3), *Peptoniphilus* spp. (n = 2), and *Atopobium parvulum*, *Anaerococcus vaginalis*, and *Porphyromonas uenonis* (n = 1 each). Table 2 shows the distribution of anaerobic microorganisms isolated from breast abscesses over the study period. 

### 2.3. Antimicrobial Susceptibility

Table 3 exhibits the antimicrobial susceptibility results for anaerobic bacteria. The most frequently isolated anaerobic group was Gram-positive anaerobic cocci (GPACs) (n = 16), especially *F. magna* (n = 11). Resistance to clindamycin was found for 56%, resistance to moxifloxacin for 31%, and resistance to metronidazole for 12%. Only 6% of isolates were resistant to penicillin. The second most frequently isolated anaerobic group was Gram-positive anaerobic bacilli (GPABs) (n = 13), with *Propionibacterium* spp. (n = 6) and *Actinomyces* spp. (n = 6) as the most common genera. Resistance to metronidazole was found for 77% and for clindamycin for 38%. Only 8% of isolates were resistant to moxifloxacin. Finally, among Gram-negative anaerobic bacilli (GNABs) (n = 6), in which *Prevotella* spp. was the most frequently isolated genus (n = 5), resistance to metronidazole and moxifloxacin was found for 50%, and to penicillin for a third of strains. Sixteen percent of isolates were resistant to clindamycin. In all groups, no resistance for imipenem, amoxicillin–clavulanic acid, and piperacillin–tazobactam was observed.

## 3. Discussion

There are few studies focused on microbiologic and clinical characteristics of breast abscesses caused only by anaerobic microorganisms. Here, we reported the antimicrobial susceptibility and clinical characteristics of 35 clinically relevant anaerobic bacteria isolated from breast abscess samples. Although staphylococci are the main causative agents in this clinical entity, anaerobic bacteria may play a role in some circumstances. Over the study period, in our case, 49 aerobic microorganisms (mainly *S. aureus*) were isolated from breast abscesses (65%), whereas 35 anaerobic pathogens were isolated (35%). This result is higher than those obtained in other previously published studies [6,7] and in our own department 25 years ago [8]. Moreover, other authors found that the anaerobes were more frequently encountered in recurring breast abscesses [9,10]. Only two cases in our series relapsed from a primary abscess. A study recovered anaerobic microorganisms twice as often as aerobes from acute infections, but the number was similar from chronic infections [4]. A study found a relationship between the size of the abscess and the etiology; smaller abscesses (≤3.0 cm) predominantly had mixed anaerobic pathogens [11]. However, in our case, the study did not investigate the relationship between the diameter of the abscess and the type of bacterial flora.

Regarding the anaerobic species isolated, the majority of previously reported studies found *Finegoldia magna* (formerly *Peptostreptococcus magnus*), *Anaerococcus prevotii* (formerly *Peptostreptococcus prevotii*), and *Parvimonas micra* (formerly *Peptostreptococcus micros*) as the main etiologic agents of this entity [3,4,8]. Moreover, in some studies, *Bacteroides* spp. and other Gram-negative anaerobic bacilli were found as the cause of breast abscesses [3,8]. In a study, *B. fragilis* represents a third of all cases of breast abscesses [7]. However, in another study, only two anaerobic cultures were positive for *Propionibacterium acnes* and *Peptostreptococcus anaerobius* [12]. In the present report, we mainly isolated *F. magna*, *Actinomyces* spp., and *Propionibacterium* spp., followed by *Prevotella* spp. and *Peptoniphilus* spp. Surprisingly, no cases of *Bacteroides* spp. were observed in our series. No special situations or indicators have demonstrated greater recovery of anaerobic microorganisms from breast abscesses. These different data between studies highlight the need for obtaining a good sample and performing subsequent culture (including in anaerobic atmosphere) and antimicrobial susceptibility testing to establish the best therapeutic strategy. In this sense, drainage of the abscess by fine needle aspiration or performing surgical drainage seems to be the best diagnostic method. 

There are few data on the risk factors associated with development of primary breast abscesses and recurrences. However, some of them seem to be implicated for development of this disease, especially tobacco smoking [13]. Furthermore, nipple piercing is associated with increased risk of developing subareolar breast abscess [13]. In some cases, subareolar abscesses have been linked to squamous metaplasia of the lactiferous ducts [14]. Although in the present report, the number of breast abscesses was not very high, six patients (25%) with breast abscess were long-term smokers; moreover, eight women (33%) suffered several episodes of inflammatory mastitis prior to the appearance of the abscess. Other patients in this series had diabetes mellitus, obesity, breast surgery, and fibrocystic mastopathy. Only two patients were previously diagnosed with breast cancer. More studies are needed to demonstrate a true relationship between these underlying conditions and the presence of breast abscess. 

Regarding the antimicrobial susceptibility of these strains, this study also showed the percentage of resistance of these anaerobic microorganisms. GPACs showed a high level of resistance to clindamycin and moxifloxacin. By contrast, GPABs showed a high level of resistance to metronidazole and clindamycin. Finally, GNABs demonstrated a high level of resistance to metronidazole, moxifloxacin, and benzylpenicillin. Overall, no resistance was observed to imipenem, piperacillin–tazobactam, and amoxicillin–clavulanic acid. Overall, antimicrobial resistance among anaerobic bacteria is increasing globally [15], and various rates of antibiotic resistance to clindamycin have been demonstrated [16,17,18,19,20]. Clindamycin resistance rates among GPACs range from 7% to 20% and are increasing especially in *F. magna* and *Peptoniphilus* species [15]. A recent study showed a clindamycin resistance rate of 25% among GPACs and 28% among *F. magna* [19]. In our study, six strains of *F. magna* (54.5%) and three of *Peptoniphilus* (75%) were resistant to clindamycin. Two of them were previously published [21,22]. By contrast, three isolates of *Actinomyces* (50%) and two of *Propionibacterium* (33%) were also resistant to this drug. Only one isolate of *Prevotella* (20%) was resistant to clindamycin. 

Reviewing the medical literature, there are not many data on antimicrobial susceptibility in anaerobes isolated from breast abscesses. The majority of studies do not provide clear information about patterns of susceptibility [4,6,8,10,11,12]. However, a study showed the antimicrobial susceptibility patterns of anaerobic bacterial isolates from breast abscess infections [7]. They found that metronidazole, amoxicillin–clavulanic acid, and piperacillin–tazobactam had 100% activity against anaerobes, similar to the present report. Some differences could be observed because they found 100% activity of clindamycin against anaerobic streptococci, while in our study, 44% of GPAC strains were resistant to this drug. By contrast, they found 100% activity of anaerobic streptococci against penicillin G, but in our report, the activity was 94%. Another study showed a high level of susceptibility to metronidazole and clindamycin to GPACs [3], which was opposite to our results. As can be seen, antimicrobial susceptibility testing of anaerobic bacteria is performed by a minority of laboratories, and although current recommendations emphasize that it is only needed for severe infections, the increase of anaerobe resistance to some antimicrobial agents indicates a greater need for this testing [23]. The main inconvenience of this study was the relatively low incidence of anaerobic microorganisms as cause of breast abscesses, so it does not allow conclusive results. 

## 4. Materials and Methods

### 4.1. Clinical Features

This was a retrospective study of microbiological data obtained from our laboratory information system (LIS). Data were gathered on age, localization, infection type (puerperal vs. non-puerperal), underlying diseases or conditions, clinical manifestations, diagnostic method, treatment, and outcome.

### 4.2. Bacterial Origin

This study was performed in the Department of Microbiology of a 700-bed university hospital in Granada (Spain). Samples from women with suspected breast abscess were included in this work from March 2017 to February 2020.

### 4.3. Isolation and Identification of Strains

All clinical samples were inoculated onto aerobic and anaerobic blood agar (BD Columbia Agar 5% Sheep Blood, Becton Dickinson, Franklin Lakes, NY), chocolate agar (BD Choco Agar, Becton Dickinson), mannitol agar (BD Mannitol Salt, Becton Dickinson), Mac Conkey agar (BD Mac Conkey II, Becton Dickinson), and thioglycolate broth (BD Fluid Thioglycolate Medium, Becton Dickinson), incubating all media at 35–37 °C for 5 days. Anaerobic plates were incubated in an anaerobic atmosphere generated with the AnaeroGen Compact anaerobic system (Oxoid Ltd., Wide Road, Basingstoke, England) at 35–37 °C. Identification of all isolates was carried out using MALDI-TOF MS (Bruker Biotyper, Bellerica, MA, USA), following the manufacturer’s recommendations. Only strains with a log (score) ≥2.0 were included and interpreted as high confidence [24] and identified only with this technique. Two isolates with the lowest log (score) were further identified by 16S rRNA gene sequencing [25].

### 4.4. Antimicrobial Susceptibility Testing

Antimicrobial susceptibility testing was performed with the gradient diffusion method using Etests (bioMérieux, Marcy l’Etoile, France) against seven antibiotics: benzyl-penicillin, amoxicillin–clavulanic acid, piperacillin–tazobactam, imipenem, moxifloxacin, clindamycin, and metronidazole. The method was performed according to Clinical and Laboratory Standards Institute (CLSI) Standard M11 A8 [26]. All anaerobic strains were sub-cultured in *Brucella* agar supplemented with 5% laked sheep blood, hemin, and 10 µg/mL vitamin K1 (BD, Becton Dickinson), and plates were incubated in anaerobic atmosphere at 35–37 °C for 48 h. Antimicrobial susceptibility results were interpreted as “susceptible”, “resistant” or “susceptible, or increased exposure” (formerly “intermediate”), according to European Committee on Antimicrobial Susceptibility Testing (EUCAST) breakpoints, except for moxifloxacin (CLSI breakpoints) [27,28].

## 5. Conclusions

This report demonstrated that anaerobic microorganisms are also involved in the etiology of breast abscesses. Overall, GPACs are the main anaerobic bacteria encountered in clinical samples obtained from these entities. Moreover, high rates of clindamycin resistance could be observed in anaerobic isolates, especially in GPACs. Although staphylococci are the main causative agents in this clinical entity, searching for anaerobes should not be forgotten, because they may cause a not inconsiderable percentage of this disease.

## Figures and Tables

**Table 1 antibiotics-09-00341-t001:** Characteristics of 24 patients with breast abscesses caused by anaerobes.

Characteristics	n (%)
**Clinical findings ***	
Abscess signs	14 (58.3)
Breast pain	10 (41.6)
Nodulation	8 (33.3)
Fever	5 (20.8)
**Underlying diseases or conditions ***	
Repetitive inflammatory mastitis	8 (33.3)
Smoker	6 (25)
DM, obesity, breast surgery	5 (20.8)
Breast cancer	2 (8.3)
Fibrocystic mastopathy	1 (4.1)
None	7 (29.1)
**Type**	
Non-puerperal	20 (90.9)
Puerperal	2 (9.1)
**Diagnostic method**	
Surgical drainage	23 (95.8)
Swab	1 (4.1)
**Localization**	
Right breast	13 (54.1)
Left breast	11 (45.9)
**Empiric treatment**	
One drug	22 (91.6)
Amoxicillin-clavulanic acid	12
Clindamycin	7
More than one drug	2 (8.4)
**Outcome**	
Favorable	22 (91.6)
Relapse	2 (8.4)

* Some patients had more than one symptom and/or underlying condition.

**Table 2 antibiotics-09-00341-t002:** Anaerobic microorganisms isolated from breast abscesses.

Microorganism	Pure Culture	Mixed Culture
	(14/40%)	(21/60%)
***Finegoldia magna***	7	4
***Prevotella***	2	3
*P. buccae*	1	0
*P. bivia*	1	2
*P. bergensis*	0	1
***Peptoniphilus***	2	2
*P. harei*	2	0
*P. gorbachii*	0	2
***Actinomyces***	1	5
*A. turicensis*	1	1
*A. radingae*	0	2
*A. neuii*	0	1
*A. europaeus*	0	1
***Propionibacterium***	2	4
*P. acnes*	2	1
*P. avidum*	0	3
***Atopobium parvulum***	0	1
***Anaerococcus vaginalis***	0	1
***Porphyromonas uenonis***	0	1

**Table 3 antibiotics-09-00341-t003:** Resistance rate (%) of anaerobic bacteria against selected antimicrobial agents obtained from breast abscesses.

	Number	BEN	MET	MOX	IMI	AMC	CLI	PIT
**GPACs**	**16**	6.2	12.5	31.2	0	0	56.2	0
**GPABs**	**13**	0	76.9	7.6	0	0	38.4	0
**GNABs**	**6**	33.3	50	50	0	0	16.6	0

GPACs: Gram-positive anaerobic cocci; GPABs: Gram-positive anaerobic bacilli; GNABs: Gram-negative anaerobic bacilli; BEN: benzylpenicillin; MET: metronidazole; MOX: moxifloxacin; IMI: imipenem; MER: meropenem; AMC: amoxicillin–clavulanic acid; CLI: clindamycin; PIT: piperacillin–tazobactam; VAN: vancomycin.

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
