# Peer review of "Breast Abscesses Caused by Anaerobic Microorganisms: Clinical and Microbiological Characteristics"

_antibiotics, 2020, doi:10.3390/antibiotics9060341_

Round 1
Reviewer 1 Report
Work of low clinical and cognitive value (read the conclusions). For a long time it has been recommended to limit elective antibiotic therapy where possible. The gold standard is the incision of an abscess with drainage or a biopsy under ultrasound guidance, e.g. an inflammatory breast cyst. Selective antibiotic therapy is permissible in exceptional situations, e.g. lack of consent for surgery, serious comorbidities (e.g. diabetes), cancer, (chemical treatment), and immune deficits. Thus, targeted antibiotic therapy may be acceptable. In general, abscess culture does not change the way it is done. Aim of the study: "treatment results", among others, does not correspond to the endpoint.
Author Response
Responses to Reviewer #1:
1- Work of low clinical and cognitive value.
R: we strongly disagree with this reviewer, because there is not much articles focused in the study of only anaerobic microorganisms in breast abscesses, so from our point of view this work could be interesting.
2- For a long time it has been recommended to limit elective antibiotic therapy where possible. Selective antibiotic therapy is permissible in exceptional situations………..and immune deficits.
R: However, appropriate antibiotic choice is an important consideration now and adjunctive antimicrobial drug therapy is frequently necessary.
3- The gold standard is the incision of an abscess with drainage or a biopsy……..breast cyst.
R: yes. In fact, in our series, incision and drainage of the abscess was performed in 95.8% of patients. In the discussion section, we added this statement.
Reviewer 2 Report
The manuscript by Cobo et al., aims to describe the anaerobic microbes present in breast abscesses and to characterize the antibiotic resistance profile of said microbes. This paper utilizes a modest size panel of less than 40 isolates collected at one hospital. Bacteria were cultured and characterized using standard methodology. The results highlight the involvement of anaerobic bacteria in these abscesses. However, the analysis f results could be greatly expanded to make more impactful conclusions. This paper will be of interest to those treating these types of infections. This is a generally well written paper that with expansion of the analysis and discussion, could be of significant interest to this subgroup. I recommend acceptance pending expansion of the discussion of the following topics:
The authors provide no information as to the mix of organisms found in these abscesses. Given that Staphylocccus is a common aerobic component, this needs to be more fully addresses. Were all of the abscesses mixed cultures? If so, were any strictly anaerobic organisms? Did the presence of anaerobic organisms at all correlate with the size or severity of the abscess? How should we distinguish between potential pathogens and the normal microbiome? Given that anaerobic cultures are not always performed, are there situations or indicators that the clinician should be aware of that would indicate the needs to investigate the presence of anaerobic bacteria?
Over 90% of these isolates were from non-puerperal women. Were there differences seen in the puerperal isolates?
There have been numerous studies done recently characterizing the microbiome of the breast, and it’s changes over the course of pregnancy and the puerperal period. Do any of these changes correlate with the presence of these anaerobic bacteria that were found?
Are these rates of antibiotic resistance unusual for these organisms? Clindamycin resistance is often associated with the accumulation of mutations, rather than acquisition of new genes. Therefore I would like more information on how common resistance to these drugs are in these species. Also, how does this compare to resistance in any of the co-cultures aerobic bacteria? Is this a case of the entire mixed culture being resistant, or only certain species?
Author Response
Reviewer #2:
1- The authors provide no information as to the mix of organisms found in these abscesses. Were all of the abscesses mixed cultures?
R: No. From 24 abscesses included, only 10 abscesses caused by anaerobes were mixed cultures; the remaining (14) were in pure culture. Please, see the “Isolated bacteria” section.
2- Were any strictly anaerobic organisms?
R: All the microorganisms here included were strict anaerobes, and their growth was only observed in the anaerobic plates.
3- Did the presence of anaerobic organisms at all correlate with the size of severity of the abscesses?
R: No. After reviewing the medical records, different size or severity could not be observed with the aerobic microorganisms. We have added a statement for this in the line 13-14 of the discussion section.
4- How should we distinguish between potential pathogens and the normal microbiome?
R: it is true that several microorganisms such as Propionibacterium, Actinomyces, and others could be present as the normal microbiota in the skin. In order to distinguish a pathogen of a contaminant, microbiological and clinical aspects were taked into account. In this sense, we have added a statement clarifying it in lines 2-3 of the “Isolated bacteria” section.
5- Are there situations or indicators that the clinician should be aware of that would indicate the needs to investigate the presence of anaerobic bacteria?
R: To our best of knowledge, no indicators have demonstrated higher rate of success for obtaining anaerobic microorganisms in breast abscesses culture. We have found only a study (reference 11) in which a correlation between smaller abscesses and mixed anaerobic pathogens could be observed. Thus, we strongly recommend the sampling and culture of all breast abscesses including in anaerobic atmosphere. We have added a statement regarding to this in the second paragraph, lines 10-12, in the discussion section.
6- Were there differences seen in the puerperal isolates?
R: Unfortunately, only two breast abscesses were diagnosed in puerperal women, so no differences could be seen. The abscesses in puerperal women were as follows: pure culture with P. acnes in one patient, and a mixed culture with Propionibacterium avidum and Prevotella bivia in the other one.
7- There have been numerous……..puerperal period. Do any of these changes correlate with the presence of these anaerobic bacteria that were found?
R: To our best of knowledge, no correlation has been still observed between puerperal breast abscesses and the presence of anaerobic microorganisms. In fact, most of abscesses in this category are caused by S. aureus. Anaerobic microorganisms are present especially in non-puerperal abscesses, lonely or with mixed growth patterns.
8- Are these rates of antibiotic resistance unusual for these organisms? Clindamycin resistance is often associated with the accumulation of mutations, rather than acquisition of new genes. Therefore I would like more information on how common resistance to these drugs are in these species. Also, how does this compare to resistance in any of the co-cultures aerobic bacteria? Is this a case of the entire mixed culture being resistant, or only certain species?
R: Antimicrobial resistance among anaerobic bacteria is increasing globally. For example, some studies showed a clindamycin resistance rate that varies 3-36% in Prevotella. Clindamycin resistance rates among GPAC range from 7% to 20% and is raising especially in F. magna and Peptoniphilus species. A recent study shows a clindamycin resistance rate of 25% among among GPAC and 28% among F. magna.
In our study, only certain species were resistant. For example, 6 from 11 strains of F. magna were resistant to clindamycin; 3 from 4 of Peptoniphilus and 3 from 6 of Actinomyces. We have added some more information about resistance to clindamycin in the discussion section (fourth paragraph, 6-10 lines).
Reviewer 3 Report
The authors submitted an interesting findings about antimicrobial susceptibility of anaerobic bacteria isolates from breast abscesses. The manuscript is well written and presented. The use of MALDI-TOF MS in the study identified more isolates compared to their earlier studies.
My one concern is whether the authors confirm the isolates using a secondary biochemical assays?
Author Response
Reviewer #3:
1- My one concern is whether the authors confirm the isolates using a secondary biochemical assays?
R: the majority of isolates were only identified by MALDI-.TOF MS, because the log (score) was >2.0 and they were interpreted as high confidence. Only in two strains, an additional technique (16S rRNA gene sequencing) was used. We have added this information in the text (material and methods section, isolation and identification of isolates sub-section, last line).
Round 2
Reviewer 1 Report
In the modern medicine, due to the increasing resistance of bacteria to antibiotics, due to their widespread use (not always in targeted way), we try to limit their administration to the necessary cases only. Here the basic procedure is surgery. Thus, the work is mainly cognitive in its character, but retrospective.
I have three comments.
Introduction: last sentence - "analyzing clinical results ..." better - analyzing clinical data
Discussion: "However, in our case, no differences could be observed in this sense" - the study did not investigate the relationship between the diameter of the abscess and the type of bacterial flora.
Conclusions: the first two conclusions do not result from the work
Author Response
1- As the reviewer requested, we have better put “analyzing clinical data”.
2- Discussion: as requested, we have put: “However, in our case, the study did not investigate the relationship between the diameter of the abscess and the type of bacterial flora”.
3- We have rewritten the conclusions’ section.
Reviewer 2 Report
The authors do not substantive address my above claims. The addition of the statement "microbiological and clinical aspects were taked into account." contains a typo and does not substantively address how these decisions were made. The authors need to take the time to read more of the literature on the issues mentioned by reviewers and utilize those sources in their discussion with citations.
I do not feel that these revisions are sufficient for acceptance.
Author Response
1- The authors do not substantive address my above claims. The addition of the statement "microbiological and clinical aspects were taked into account." contains a typo and does not substantively address how these decisions were made. The authors need to take the time to read more of the literature on the issues mentioned by reviewers and utilize those sources in their discussion with citations.
I do not feel that these revisions are sufficient for acceptance.
R: In order to distinguish true pathogens and contaminants, microbiological and clinical aspects were taked into account. The presence of abscess infection was defined as an episode of erythematous pain, localized edema with tenderness on palpation, and/or signs of suppuration in any breast location [4]. Patients with a history of at least two recurrent episodes and/or presentation with draining sinuses were also included [4]. On the other hand, puerperal breast abscesses were defined as breast abscesses during pregnancy, during the first 3 months after labor without lactation, during lactation, or during the first 3 months after cessation of lactation [5]. From a microbiological point of view, isolates in pure culture with abundant growth in the plates were included. Also, mixed cultures with presence of abundant polymorphonuclear leukocytes were considered.